# Deep Character-Level Neural Machine Translation By Learning Morphology

**Shenjian Zhao**
Department of Computer Science and Engineering
Shanghai Jiao Tong University
Shanghai 200240, China
`sword.york@gmail.com`

**Zhihua Zhang**
School of Mathematical Sciences
Peking University
Beijing 100871, China
`zhzhang@math.pku.edu.cn`

## Abstract

Neural machine translation aims at building a single large neural network that can be trained to maximize translation performance. The encoder-decoder architecture with an attention mechanism achieves a translation performance comparable to the existing state-of-the-art phrase-based systems. However, the use of large vocabulary becomes the bottleneck in both training and improving the performance. In this paper, we propose a novel architecture which learns morphology by using two recurrent networks and a hierarchical decoder which translates at character level. This gives rise to a deep character-level model consisting of six recurrent networks. Such a deep model has two major advantages. It avoids the large vocabulary issue radically; at the same time, it is more efficient in training than word-based models. Our model obtains a higher BLEU score than the bpe-based model after training for one epoch on En-Fr and En-Cs translation tasks. Further analyses show that our model is able to learn morphology.

## 1 Introduction

Neural machine translation (NMT) attempts to build a single large neural network that reads a sentence and outputs a translation (Sutskever et al., 2014). Most of the extant neural machine translations models belong to a family of word-level encoder-decoders (Sutskever et al., 2014; Cho et al., 2014). Recently, Bahdanau et al. (2015) proposed a model with attention mechanism which automatically searches the alignments and greatly improves the performance. However, the use of a large vocabulary seems necessary for the word-level neural machine translation models to improve performance (Sutskever et al., 2014; Cho et al., 2015).

Chung et al. (2016a) listed three reasons behind the wide adoption of word-level modeling: (i) word is a basic unit of a language, (ii) data sparsity, (iii) vanishing gradient of character-level modeling. Consider that a language itself is an evolving system. So it is impossible to cover all words in the language. The problem of rare words that are out of vocabulary (OOV) is a critical issue which can effect the performance of neural machine translation. In particular, using larger vocabulary does improve performance (Sutskever et al., 2014; Cho et al., 2015). However, the training becomes much harder and the vocabulary is often filled with many similar words that share a lexeme but have different morphology.

There are many approaches to dealing with the out-of-vocabulary issue. For example, Gulcehre et al. (2016); Luong et al. (2015); Cho et al. (2015) proposed to obtain the alignment information of target unknown words, after which simple word dictionary lookup or identity copy can be performed to replace the unknown words in translation. However, these approaches ignore several important properties of languages such as monolinguality and crosslinguality as pointed out by Luong and

Manning (2016). Thus, Luong and Manning (2016) proposed a hybrid neural machine translation model which leverages the power of both words and characters to achieve the goal of open vocabulary neural machine translation.

Intuitively, it is elegant to directly model pure characters. However, as the length of sequence grows significantly, character-level translation models have failed to produce competitive results compared with word-based models. In addition, they require more memory and computation resource. Especially, it is much difficult to train the attention component. For example, Ling et al. (2015a) proposed a compositional character to word (C2W) model and applied it to machine translation (Ling et al., 2015b). They also used a hierarchical decoder which has been explored before in other context (Serban et al., 2015). However, they found it slow and difficult to train the character-level models, and one has to resort to layer-wise training the neural network and applying supervision for the attention component. In fact, such RNNs often struggle with separating words that have similar morphologies but very different meanings.

In order to address the issues mentioned earlier, we introduce a novel architecture by exploiting the structure of words. It is built on two recurrent neural networks: one for learning the representation of preceding characters and another for learning the weight of this representation of the whole word. Unlike subword-level model based on the byte pair encoding (BPE) algorithm (Sennrich et al., 2016), we learn the subword unit automatically. Compared with CNN word encoder (Kim et al., 2016; Lee et al., 2016), our model is able to generate a meaningful representation of the word. To decode at character level, we devise a hierarchical decoder which sets the state of the second-level RNN (character-level decoder) to the output of the first-level RNN (word-level decoder), which will generate a character sequence until generating a delimiter. In this way, our model almost keeps the same encoding length for encoder as word-based models but eliminates the use of a large vocabulary. Furthermore, we are able to efficiently train the deep model which consists of six recurrent networks, achieving higher performance.

In summary, we propose a hierarchical architecture (character -> subword -> word -> source sentence -> target word -> target character) to train a deep character-level neural machine translator. We show that the model achieves a high translation performance which is comparable to the state-of-the-art neural machine translation model on the task of En-Fr, En-Cs and Cs-En translation. The experiments and analyses further support the statement that our model is able to learn the morphology.

## 2 NEURAL MACHINE TRANSLATION

Neural machine translation is often implemented as an encoder-decoder architecture. The encoder usually uses a recurrent neural network (RNN) or a bidirectional recurrent neural network (BiRNN) (Schuster and Paliwal, 1997) to encode the input sentence $\mathbf{x} = \{x_1, \ldots, x_{T_x}\}$ into a sequence of hidden states $\mathbf{h} = \{\mathbf{h}_1, \ldots, \mathbf{h}_{T_x}\}$:

$$\mathbf{h}_t = f_1(\mathbf{e}(x_t), \mathbf{h}_{t-1}),$$

where $\mathbf{e}(x_t) \in \mathbb{R}^m$ is an $m$-dimensional embedding of $x_t$. The decoder, another RNN, is often trained to predict next word $y_t$ given previous predicted words $\{y_1, \ldots, y_{t-1}\}$ and the context vector $\mathbf{c}_t$; that is,

$$p(y_t \mid \{y_1, \ldots, y_{t-1}\}) = g(\mathbf{e}(y_{t-1}), \mathbf{s}_t, \mathbf{c}_t),$$

where

$$\mathbf{s}_t = f_2(\mathbf{e}(y_{t-1}), \mathbf{s}_{t-1}, \mathbf{c}_t) \tag{1}$$

and $g$ is a nonlinear and potentially multi-layered function that computes the probability of $y_t$. The context $\mathbf{c}_t$ depends on the sequence of $\{\mathbf{h}_1, \ldots, \mathbf{h}_{T_x}\}$. Sutskever et al. (2014) encoded all information in the source sentence into a fixed-length vector, i.e., $\mathbf{c}_t = \mathbf{h}_{T_x}$. Bahdanau et al. (2015) computed $\mathbf{c}_t$ by the alignment model which handles the bottleneck that the former approach meets.

The whole model is jointly trained by maximizing the conditional log-probability of the correct translation given a source sentence with respect to the parameters of the model $\boldsymbol{\theta}$:

$$\boldsymbol{\theta}^* = \underset{\boldsymbol{\theta}}{\mathrm{argmax}} \sum_{t=1}^{T_y} \log p(y_t \mid \{y_1, \ldots, y_{t-1}\}, \mathbf{x}, \boldsymbol{\theta}).$$

For the detailed description of the implementation, we refer the reader to the papers (Sutskever et al., 2014; Bahdanau et al., 2015).

## 3    DEEP CHARACTER-LEVEL NEURAL MACHINE TRANSLATION

We consider two problems in the word-level neural machine translation models. First, how can we map a word to a vector? It is usually done by a lookup table (embedding matrix) where the size of vocabulary is limited. Second, how do we map a vector to a word when predicting? It is usually done via a softmax function. However, the large vocabulary will make the softmax intractable computationally.

We correspondingly devise two novel architectures, a word encoder which utilizes the morphology and a hierarchical decoder which decodes at character level. Accordingly, we propose a deep character-level neural machine translation model (DCNMT).

### 3.1    LEARNING MORPHOLOGY IN A WORD ENCODER

Many words can be subdivided into smaller meaningful units called morphemes, such as "any-one", "any-thing" and "every-one." At the basic level, words are made of morphemes which are recognized as grammatically significant or meaningful. Different combinations of morphemes lead to different meanings. Based on these facts, we introduce a word encoder to learn the morphemes and the rules of how they are combined. Even if the word encoder had never seen "everything" before, with a understanding of English morphology, the word encoder could gather the meaning easily. Thus learning morphology in a word encoder might speedup training.

The word encoder is based on two recurrent neural networks, as illustrated in Figure 1. We compute the representation of the word 'anyone' as

$$\mathbf{r}_{\text{anyone}} = \tanh(\sum_{t=1}^{6} w_t \mathbf{r}_t),$$

where $\mathbf{r}_t$ is an RNN hidden state at time $t$, computed by

$$\mathbf{r}_t = f(\mathbf{e}(x_t), \mathbf{r}_{t-1}).$$

Each $\mathbf{r}_t$ contains information about the preceding characters. The weight $w_t$ of each representation $\mathbf{r}_t$ is computed by

$$w_t = \exp(\text{aff}(\mathbf{h}_t)),$$

where $\mathbf{h}_t$ is another RNN hidden state at time $t$ and aff() is an affine function which maps $\mathbf{h}_t$ to a scalar. Here, we use a BiRNN to compute $\mathbf{h}_t$ as shown in Figure 1. Instead of normalizing it by $\sum_t \exp(\text{aff}(\mathbf{h}_t))$, we use an activation function $\tanh$ as it performs best in experiments.

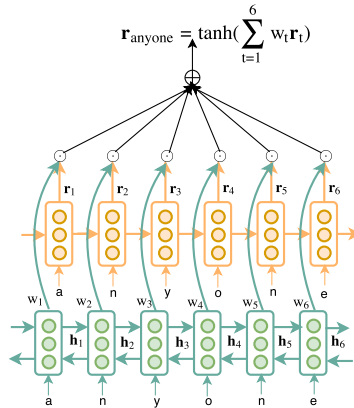

Figure 1: The representation of the word 'anyone.'

We can regard the weight $w_i$ as the energy that determines whether $\mathbf{r}_i$ is a representation of a morpheme and how it contributes to the representation of the word. Compared with an embedding lookup table, the decoupled RNNs learn the representation of morphemes and the rules of how they are combined respectively, which may be viewed as learning distributed representations of words explicitly. For example, we are able to translate "convenienter" correctly which validates our idea.

After obtaining the representation of the word, we could encode the sentence using a bidirectional RNN as RNNsearch (Bahdanau et al., 2015). The detailed architecture is shown in Figure 2.

### 3.2    HIERARCHICAL DECODER

To decode at the character level, we introduce a hierarchical decoder. The first-level decoder is similar to RNNsearch which contains the information of the target word. Specifically, $\mathbf{s}_t$ in Eqn. (1) contains the information of target word at time $t$. Instead of using a multi-layer network following a softmax function to compute the probability of each target word using $\mathbf{s}_t$, we employ a second-level decoder which generates a character sequence based on $\mathbf{s}_t$.

We proposed a variant of the gate recurrent unit (GRU) (Cho et al., 2014; Chung et al., 2014) that used in the second-level decoder and we denote it as HGRU (It is possible to use the LSTM (Hochreiter

and Schmidhuber, 1997) units instead of the GRU described here). HGRU has a settable state and generates character sequence based on the given state until generating a delimiter. In our model, the state is initialized by the output of the first-level decoder. Once HGRU generates a delimiter, it will set the state to the next output of the first-level decoder. Given the previous output character sequence $\{y_0, y_1, \ldots, y_{t-1}\}$ where $y_0$ is a token representing the start of sentence, and the auxiliary sequence $\{a_0, a_1, \ldots, a_{t-1}\}$ which only contains 0 and 1 to indicate whether $y_i$ is a delimiter ($a_0$ is set to 1), HGRU updates the state as follows:

$$\mathbf{g}_{t-1} = (1 - a_{t-1})\mathbf{g}_{t-1} + a_{t-1}\mathbf{s}_{i_t}, \tag{2}$$

$$\mathbf{q}_t^j = \sigma([\mathbf{W}_q\mathbf{e}(y_{t-1})]^j + [\mathbf{U}_q\mathbf{g}_{t-1}]^j), \tag{3}$$

$$\mathbf{z}_t^j = \sigma([\mathbf{W}_z\mathbf{e}(y_{t-1})]^j + [\mathbf{U}_z\mathbf{g}_{t-1}]^j), \tag{4}$$

$$\tilde{\mathbf{g}}_t^j = \phi([\mathbf{W}\mathbf{e}(y_{t-1})]^j + [\mathbf{U}(\mathbf{q}_t \odot \mathbf{g}_{t-1})]^j), \tag{5}$$

$$\mathbf{g}_t^j = \mathbf{z}_t^j\mathbf{g}_{t-1}^j + (1 - \mathbf{z}_t^j)\tilde{\mathbf{g}}_t^j, \tag{6}$$

where $\mathbf{s}_{i_t}$ is the output of the first-level decoder which calculated as Eqn. (8). We can compute the probability of each target character $y_t$ based on $\mathbf{g}_t$ with a softmax function:

$$p(y_t \mid \{y_1, \ldots, y_{t-1}\}, \mathbf{x}) = \text{softmax}(\mathbf{g}_t). \tag{7}$$

The current problem is that the number of outputs of the first-level decoder is much fewer than the target character sequence. It will be intractable to conditionally pick outputs from the the first-level decoder when training in batch manner (at least intractable for Theano (Bastien et al., 2012) and other symbolic deep learning frameworks to build symbolic expressions). Luong and Manning (2016) uses two forward passes (one for word-level and another for character-level) in batch training which is less efficient. However, in our model, we use a matrix to unfold the outputs of the first-level decoder, which makes the batch training process more efficient. It is a $T_y \times T$ matrix $\mathbf{R}$, where $T_y$ is the number of delimiter (number of words) in the target character sequence and $T$ is the length of the target character sequence. $\mathbf{R}[i, j_1 + 1]$ to $\mathbf{R}[i, j_2]$ are set as 1 if $j_1$ is the index of the $(i-1)$-th delimiter and $j_2$ is the index of the $i$-th delimiter in the target character sequence. The index of the 0-th delimiter is set as 0. For example, when the target output is "g o ! " and the output of the first-level decoder is $[\mathbf{s}_1, \mathbf{s}_2]$, the unfolding step will be:

$$[\mathbf{s}_1, \mathbf{s}_2] \begin{bmatrix} 1 & 1 & 1 & 0 & 0 \\ 0 & 0 & 0 & 1 & 1 \end{bmatrix} = [\mathbf{s}_1, \mathbf{s}_1, \mathbf{s}_1, \mathbf{s}_2, \mathbf{s}_2],$$

therefore $\{\mathbf{s}_{i_1}, \mathbf{s}_{i_2}, \mathbf{s}_{i_3}, \mathbf{s}_{i_4}, \mathbf{s}_{i_5}\}$ is correspondingly set to $\{\mathbf{s}_1, \mathbf{s}_1, \mathbf{s}_1, \mathbf{s}_2, \mathbf{s}_2\}$ in HGRU iterations. After this procedure, we can compute the probability of each target character by the second-level decoder according to Eqns. (2) to (7).

## 3.3 MODEL ARCHITECTURES

There are totally six recurrent neural networks in our model, which can be divided into four layers as shown in Figure 2. Figure 2 illustrates the training procedure of a basic deep character-level neural machine translation. It is possible to use multi-layer recurrent neural networks to make the model deeper. The first layer is a source word encoder which contains two RNNs as shown in Figure 1. The second layer is a bidirectional RNN sentence encoder which is identical to that of (Bahdanau et al., 2015). The third layer is the first-level decoder. It takes the representation of previous target word as a feedback, which is produced by the target word encoder in our model. As the feedback is less important, we use an ordinary RNN to encode the target word. The feedback $\mathbf{r}_{Y_{t-1}}$ then combines the previous hidden state $\mathbf{u}_{t-1}$ and the context $\mathbf{c}_t$ from the sentence encoder to generate the vector $\mathbf{s}_t$:

$$\mathbf{s}_t = \mathbf{W}_1\mathbf{c}_t + \mathbf{W}_2\mathbf{r}_{Y_{t-1}} + \mathbf{W}_3\mathbf{u}_{t-1} + \mathbf{b}. \tag{8}$$

With the state of HGRU in the second-level decoder setting to $\mathbf{s}_t$ and the information of previous generated character, the second-level decoder generates the next character until generating an end of sentence token (denoted as </s> in Figure 2). With such a hierarchical architecture, we can train our character-level neural translation model perfectly well in an end-to-end fashion.

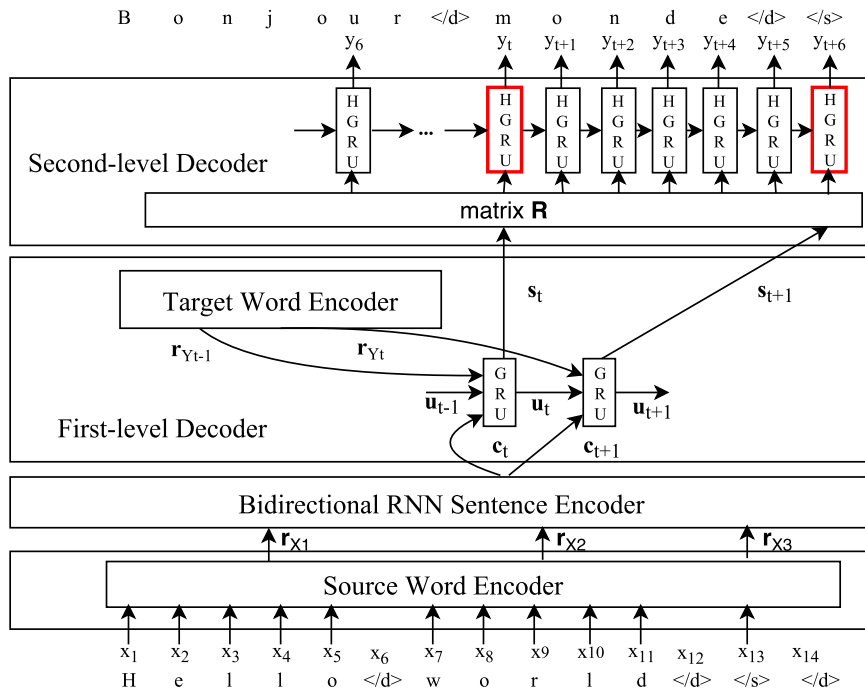

Figure 2: Deep character-level neural machine translation. The HGRUs with red border indicate that the state should be set to the output of the first-level decoder.

### 3.4 GENERATION PROCEDURE

We first encode the source sequence as in the training procedure, then we generate the target sequence character by character based on the output $s_t$ of the first-level decoder. Once we generate a delimiter, we should compute next vector $s_{t+1}$ according to Eqn. (8) by combining feedback $r_{Y_t}$ from the target word encoder, the context $c_{t+1}$ from the sentence encoder and the hidden state $u_t$. The generation procedure will terminate once an end of sentence (EOS) token is produced.

## 4 EXPERIMENTS

We implement the model using Theano (Bergstra et al., 2010; Bastien et al., 2012) and Blocks (van Merriënboer et al., 2015), the source code and the trained models are available at github [1]. We train our model on a single GTX Titan X with 12GB RAM. First we evaluate our model on English-to-French translation task where the languages are morphologically poor. For fair comparison, we use the same dataset as in RNNsearch which is the bilingual, parallel corpora provided by ACL WMT'14. In order to show the strengths of our model, we conduct on the English-to-Czech and Czech-to-English translation tasks where Czech is a morphologically rich language. We use the same dataset as (Chung et al., 2016a; Lee et al., 2016) which is provided by ACL WMT'15 [2].

### 4.1 DATASET

We use the parallel corpora for two language pairs from WMT: En-Cs and En-Fr. They consist of 15.8M and 12.1M sentence pairs, respectively. In terms of preprocessing, we only apply the usual tokenization. We choose a list of 120 most frequent characters for each language which coveres nearly 100% of the training data. Those characters not included in the list are mapped to a special token

---

[1] https://github.com/SwordYork/DCNMT
[2] http://www.statmt.org/wmt15/translation-task.html

(<unk>). We use *newstest2013*(Dev) as the development set and evaluate the models on *newstest2015* (Test). We do not use any monolingual corpus.

## 4.2 TRAINING DETAILS

We follow (Bahdanau et al., 2015) to use similar hyperparameters. The bidirectional RNN sentence encoder and the hierarchical decoder both consists of two-layer RNNs, each has 1024 hidden units; We choose 120 most frequent characters for DCNMT and the character embedding dimensionality is 64. The source word is encoded into a 600-dimensional vector. The other GRUs in our model have 512 hidden units.

We use the ADAM optimizer (Kingma and Ba, 2015) with minibatch of 56 sentences to train each model (for En-Fr we use a minibatch of 72 examples). The learning rate is first set to $10^{-3}$ and then annealed to $10^{-4}$.

We use a beam search to find a translation that approximately maximizes the conditional log-probability which is a commonly used approach in neural machine translation (Sutskever et al., 2014; Bahdanau et al., 2015). In our DCNMT model, it is reasonable to search directly on character level to generate a translation.

## 5 RESULT AND ANALYSIS

We conduct comparison of quantitative results on the En-Fr, En-Cs and Cs-En translation tasks in Section 5.1. Apart from measuring translation quality, we analyze the efficiency of our model and effects of character-level modeling in more details.

## 5.1 QUANTITATIVE RESULTS

We illustrate the efficiency of the deep character-level neural machine translation by comparing with the bpe-based subword model (Sennrich et al., 2016) and other character-level models. We measure the performance by BLEU score (Papineni et al., 2002).

Table 1: BLEU scores of different models on three language pairs.

|       | Model | Size | Src | Trgt | Length | | Epochs | Days | Dev | Test |
|-------|-------|------|-----|------|--------|--------|--------|------|-----|------|
| En-Fr | bpe2bpe[1] | - | bpe | bpe | 50 | 50 | - | - | 26.91 | 29.70 |
|       | C2W[2] | $\sim 54$ M | char | char | 300 | 300 | $\sim 2.8$ | $\sim 27$ | 25.89 | 27.04 |
|       | CNMT | $\sim 52$ M | char | char | 300 | 300 | $\sim 3.8$ | $\sim 21$ | 28.19 | 29.38 |
|       | DCNMT | $\sim 54$ M | char | char | 300 | 300 | 1 | $\sim 7$ | 27.02 | 28.13 |
|       |       |      |     |      |        |        | $\sim 2.8$ | $\sim 19$ | **29.31** | **30.56** |
| En-Cs | bpe2bpe[1] | - | bpe | bpe | 50 | 50 | - | - | 15.90 | 13.84 |
|       | bpe2char[3] | - | bpe | char | 50 | 500 | - | - | - | 16.86 |
|       | char[5] | - | char | char | 600 | 600 | > 4 | $\sim 90$ | - | 17.5 |
|       | hybrid[5] | $\sim 250$ M | hybrid | hybrid | 50 | 50 | > 4 | $\sim 21$ | - | **19.6** |
|       | DCNMT | $\sim 54$ M | char | char | 450 | 450 | 1 | $\sim 5$ | 15.50 | 14.87 |
|       |       |      |     |      |        |        | $\sim 2.9$ | $\sim 15$ | **17.89** | 16.96 |
| Cs-En | bpe2bpe[1] | - | bpe | bpe | 50 | 50 | - | - | 21.24 | 20.32 |
|       | bpe2char[3] | $\sim 76$ M | bpe | char | 50 | 500 | $\sim 6.1$ | $\sim 14$ | 23.27 | 22.42 |
|       | char2char[4] | $\sim 69$ M | char | char | 450 | 450 | $\sim 7.9$ | $\sim 30$ | **23.38** | 22.46 |
|       | DCNMT | $\sim 54$ M | char | char | 450 | 450 | 1 | $\sim 5$ | 20.50 | 19.75 |
|       |       |      |     |      |        |        | $\sim 4.6$ | $\sim 22$ | 23.24 | **22.48** |

In Table 1, "Length" indicates the maximum sentence length in training (based on the number of words or characters), "Size" is the total number of parameters in the models. We report the BLEU

scores of DCNMT when trained after one epoch in the above line and the final scores in the following line. The results of other models are taken from (1)Firat et al. (2016), (3)Chung et al. (2016a), (4)Lee et al. (2016) and (5)Luong and Manning (2016) respectively, except (2) is trained according to Ling et al. (2015b). The only difference between CNMT and DCNMT is CNMT uses an ordinary RNN to encode source words (takes the last hidden state). The training time for (3) and (4) is calculated based on the training speed in (Lee et al., 2016). For each test set, the best scores among the models per language pair are bold-faced. Obviously, character-level models are better than the subword-level models, and our model is comparable to the start-of-the-art character-level models. Note that, the purely character model of (5)(Luong and Manning, 2016) took 3 months to train and yielded +0.5 BLEU points compared to our result. We have analyzed the efficiency of our decoder in Section 3.2. Besides, our model is the simplest and the smallest one in terms of the model size.

## 5.2 LEARNING MORPHOLOGY

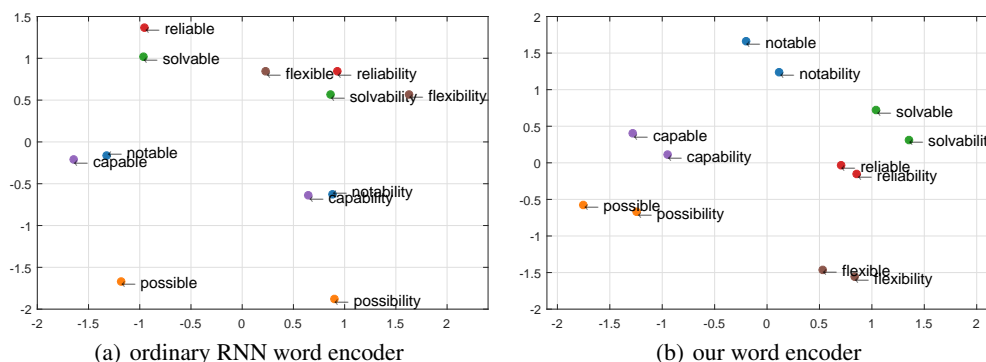

(a) ordinary RNN word encoder    (b) our word encoder

Figure 3: Two-dimensional PCA projection of the 600-dimensional representation of the words.

In this section, we investigate whether our model could learn morphology. First we want to figure out the difference between an ordinary RNN word encoder and our word encoder. We choose some words with similar meaning but different in morphology as shown in Figure 3. We could find in Figure 3(a) that the words ending with "ability", which are encoded by the ordinary RNN word encoder, are jammed together. In contrast, the representations produced by our encoder are more reasonable and the words with similar meaning are closer.

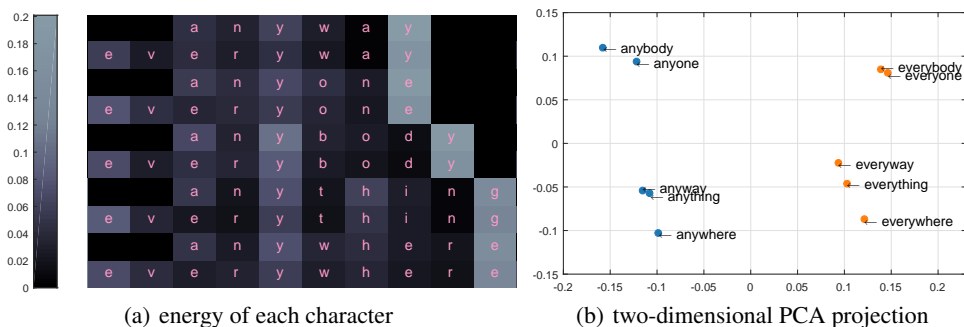

(a) energy of each character    (b) two-dimensional PCA projection

Figure 4: The learnt morphemes

Then we analyze how our word encoder learns morphemes and the rules of how they are combined. We demonstrate the encoding details on "any*" and "every*". Figure 4(a) shows the energy of each character, more precisely, the energy of preceding characters. We could see that the last character of a morpheme will result a relative large energy (weight) like "any" and "every" in these words. Moreover, even the preceding characters are different, it will produce a similar weight for the same morpheme like "way" in "anyway" and "everyway". The two-dimensional PCA projection in Figure

4(b) further validates our idea. The word encoder may be able to guess the meaning of "everything" even it had never seen "everything" before, thus speedup learning. More interestingly, we find that not only the ending letter has high energy, but also the beginning letter is important. It matches the behavior of human perception (White et al., 2008).

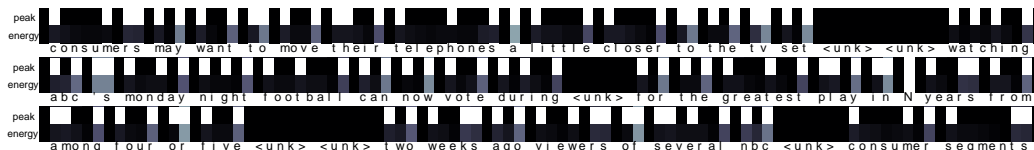

Figure 5: Subword-level boundary detected by our word encoder.

Moreover, we apply our trained word encoder to Penn Treebank Line 1. Unlike Chung et al. (2016b), we are able to detect the boundary of the subword units. As shown in Figure 5, "consumers", "monday", "football" and "greatest" are segmented into "consum-er-s","mon-day", "foot-ball" and "great-est" respectively. Since there are no explicit delimiters, it may be more difficult to detect the subword units.

### 5.3 BENEFITING FROM LEARNING MORPHOLOGY

As analyzed in Section 5.2, learning morphology could speedup learning. This has also been shown in Table 1 (En-Fr and En-Cs task) from which we see that when we train our model just for one epoch, the obtained result even outperforms the final result with bpe baseline.

Another advantage of our model is the ability to translate the misspelled words or the nonce words. The character-level model has a much better chance recovering the original word or sentence. In Table 2, we list some examples where the source sentences are taken from *newstest2013* but we change some words to misspelled words or nonce words. We also list the translations from Google translate [3] and online demo of neural machine translation by LISA.

Table 2: Sample translations.

**(a) Misspelled words**

| | |
|---|---|
| Source | For the time being *howeve* their research is *unconclusive*. |
| Reference | Leurs recherches ne sont toutefois pas concluantes pour l'instant. |
| Google translate | Pour le moment, leurs recherches ne sont *pas concluantes*. |
| LISA | Pour le moment *UNK* leur recherche est *UNK*. |
| DCNMT | Pour le moment, *cependant*, leur recherche n'est *pas concluante*. |

**(b) Nonce words (morphological change)**

| | |
|---|---|
| Source | Then we will be able to supplement the real world with virtual objects in a much *convenienter* form . |
| Reference | Ainsi , nous pourrons compléter le monde réel par des objets virtuels dans une forme *plus pratique* . |
| Google translate | Ensuite, nous serons en mesure de compléter le monde réel avec des objets virtuels dans une forme beaucoup *plus pratique*. |
| LISA | Ensuite, nous serons en mesure de compléter le vrai monde avec des objets virtuels sous une forme bien *UNK*. |
| DCNMT | Ensuite, nous serons en mesure de compléter le monde réel avec des objets virtuels dans une forme beaucoup *plus pratique*. |

As listed in Table 2(a), DCNMT is able to translate out the misspelled words correctly. For a word-based translator, it is never possible because the misspelled words are mapped into <unk>

---

[3]The translations by Google translate were made on Nov 4, 2016.

token before translating. Thus, it will produce an <unk> token or just take the word from source sentence (Gulcehre et al., 2016; Luong et al., 2015). More interestingly, DCNMT could translate "convenienter" correctly as shown in Table 2(b). By concatenating "convenient" and "er", we get the comparative adjective form of "convenient" which never appears in the training set; however, our model guessed it correctly based on the morphemes and the rules.

# 6 CONCLUSION

In this paper we have proposed an hierarchical architecture to train the deep character-level neural machine translation model by introducing a novel word encoder and a multi-leveled decoder. We have demonstrated the efficiency of the training process and the effectiveness of the model in comparison with the word-level and other character-level models. The BLEU score implies that our deep character-level neural machine translation model likely outperforms the word-level models and is competitive with the state-of-the-art character-based models. It is possible to further improve performance by using deeper recurrent networks (Wu et al., 2016), training for more epochs and training with longer sentence pairs.

As a result of the character-level modeling, we have solved the out-of-vocabulary (OOV) issue that word-level models suffer from, and we have obtained a new functionality to translate the misspelled or the nonce words. More importantly, the deep character-level is able to learn the similar embedding of the words with similar meanings like the word-level models. Finally, it would be potentially possible that the idea behind our approach could be applied to many other tasks such as speech recognition and text summarization.

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

# A   DETAILED DESCRIPTION OF THE MODEL

Here we describe the implementation using Theano, it should be applicable to other symbolic deep learning frameworks. We use $f$ to denote the transition of the recurrent network.

## A.1   SOURCE WORD ENCODER

As illustrated in Section 3.1, the word encoder is based on two recurrent neural networks. We compute the representation of the word 'anyone' as

$$\mathbf{r}_{\text{anyone}} = \tanh(\sum_{t=1}^{6} w_t \mathbf{r}_t),$$

where $\mathbf{r}_t \in \mathbb{R}^n$ is an RNN hidden state at time $t$, computed by

$$\mathbf{r}_t = f(\mathbf{e}(x_t), \mathbf{r}_{t-1}).$$

Each $\mathbf{r}_t$ contains information about the preceding characters. The weight $w_t$ of each representation $\mathbf{r}_t$ is computed by

$$w_t = \exp(\mathbf{W}_w \mathbf{h}_t + b_w),$$

where $\mathbf{W}_w \in \mathbb{R}^{1 \times 2l}$ maps the vector $\mathbf{h}_t \in \mathbb{R}^{2l}$ to a scalar and $\mathbf{h}_t$ is the state of the BiRNN at time $t$:

$$\mathbf{h}_t = \begin{bmatrix} \overrightarrow{\mathbf{h}}_t \\ \overleftarrow{\mathbf{h}}_t \end{bmatrix}. \tag{9}$$

$\overrightarrow{\mathbf{h}}_t \in \mathbb{R}^l$ is the forward state of the BiRNN which is computed by

$$\overrightarrow{\mathbf{h}}_t = f(\mathbf{e}(x_t), \overrightarrow{\mathbf{h}}_{t-1}). \tag{10}$$

The backward state $\overleftarrow{\mathbf{h}}_t \in \mathbb{R}^l$ is computed similarly, however in a reverse order.

## A.2   SOURCE SENTENCE ENCODER

After encoding the words by the source word encoder, we feed the representations to the source sentence encoder. For example, the source "Hello world </s>" is encoded into a vector $[\mathbf{r}_{\text{Hello}}, \mathbf{r}_{\text{world}}, \mathbf{r}_{</s>}]$, then the BiRNN sentence encoder encodes this vector into $[\mathbf{v}_1, \mathbf{v}_2, \mathbf{v}_3]$. The computation is the same as Eqn. (9) and Eqn. (10), however the input now changes to the representation of the words.

## A.3   FIRST-LEVEL DECODER

The first-level decoder is similar to Bahdanau et al. (2015) which utilizes the attention mechanism. Given the context vector $\mathbf{c}_t$ from encoder, the hidden state $\mathbf{u}_t \in \mathbb{R}^m$ of the GRU is computed by

$$\mathbf{u}_t = (1 - \mathbf{z}_t) \circ \mathbf{u}_{t-1} + \mathbf{z}_t \circ \tilde{\mathbf{u}}_t,$$

where

$$\tilde{\mathbf{u}}_t = \tanh(\mathbf{W} \mathbf{r}_{Y_{t-1}} + \mathbf{U}[\mathbf{q}_t \circ \mathbf{u}_{t-1}] + \mathbf{C} \mathbf{c}_t)$$
$$\mathbf{z}_t = \sigma(\mathbf{W}_z \mathbf{r}_{Y_{t-1}} + \mathbf{U}_z \mathbf{u}_{t-1} + \mathbf{C}_z \mathbf{c}_t)$$
$$\mathbf{q}_t = \sigma(\mathbf{W}_q \mathbf{r}_{Y_{t-1}} + \mathbf{U}_q \mathbf{u}_{t-1} + \mathbf{C}_q \mathbf{c}_t).$$

$\mathbf{r}_{Y_{t-1}}$ is the representation of the target word which is produced by an ordinary RNN (take the last state). The context vector $\mathbf{c}_t$ is computed by the attention mechanism at each step:

$$\mathbf{c}_t = \sum_{j=1}^{T_x} \alpha_{tj} \mathbf{v}_j,$$

where

$$\alpha_{tj} = \frac{\exp(e_{tj})}{\sum_{k=1}^{T_x} \exp(e_{tk})}$$
$$e_{tj} = \mathbf{E} \tanh(\mathbf{W}_e \mathbf{u}_{t-1} + \mathbf{H}_e \mathbf{h}_j).$$

$\mathbf{E} \in \mathbb{R}^{1 \times m}$ which maps the vector into a scalar. Then the hidden state $\mathbf{u}_t$ is further processed as Eqn. (8) before feeding to the second-level decoder:

$$\mathbf{s}_{t+1} = \mathbf{W}_1 \mathbf{c}_{t+1} + \mathbf{W}_2 \mathbf{r}_{Y_t} + \mathbf{W}_3 \mathbf{u}_t + \mathbf{b}.$$

### A.4    SECOND-LEVEL DECODER

As described in Section 3.2, the number of outputs of the first-level decoder is much fewer than the target character sequence. It will be intractable to conditionally pick outputs from the the first-level decoder when training in batch manner (at least intractable for Theano (Bastien et al., 2012) and other symbolic deep learning frameworks to build symbolic expressions). We use a matrix $\mathbf{R} \in \mathbb{R}^{T_y \times T}$ to unfold the outputs $[\mathbf{s}_1, \ldots, \mathbf{s}_{T_y}]$ of the first-level decoder ($T_y$ is the number of words in the target sentence and $T$ is the number of characters). $\mathbf{R}$ is a symbolic matrix in the final loss, it is constructed according the delimiters in the target sentences when training (see Section 3.2 for the detailed construction, note that $\mathbf{R}$ is a tensor in batch training. ). After unfolding, the input of HGRU becomes $[\mathbf{s}_{i_1}, \ldots, \mathbf{s}_{i_T}]$, that is

$$[\mathbf{s}_{i_1}, \ldots, \mathbf{s}_{i_T}] = [\mathbf{s}_1, \ldots, \mathbf{s}_{T_y}]\mathbf{R}.$$

According to Eqns.(2) to (7), we can compute the probability of each target character :

$$p(y_t \mid \{y_1, \ldots, y_{t-1}\}, \mathbf{x}) = \text{softmax}(\mathbf{g}_t).$$

Finally, we could compute the cross-entropy loss and train with SGD algorithm.

## B    SAMPLE TRANSLATIONS

We show additional sample translations in the following Tables.

Table 3: Sample translations of En-Fr.

| | |
|---|---|
| Source | This " disturbance " produces an electromagnetic wave ( of light , infrared , ultraviolet etc . ) , and this wave is nothing other than a photon - and thus one of the " force carrier " bosons . |
| Reference | Quand , en effet , une particule ayant une charge électrique accélère ou change de direction , cela " dérange " le champ électromagnétique en cet endroit précis , un peu comme un caillou lancé dans un étang . |
| DCNMT | Lorsque , en fait , une particule ayant une charge électrique accélère ou change de direction , cela " perturbe " le champ électromagnétique dans cet endroit spécifique , plutôt comme un galet jeté dans un étang . |
| Source | Since October , a manifesto , signed by palliative care luminaries including Dr Balfour Mount and Dr Bernard Lapointe , has been circulating to demonstrate their opposition to such an initiative . |
| Reference | Depuis le mois d' octobre , un manifeste , signé de sommités des soins palliatifs dont le Dr Balfour Mount et le Dr Bernard Lapointe , circule pour témoigner de leur opposition à une telle initiative . |
| DCNMT | Depuis octobre , un manifeste , signé par des liminaires de soins palliatifs , dont le Dr Balfour Mount et le Dr Bernard Lapointe , a circulé pour démontrer leur opposition à une telle initiative . |

Table 4: Sample translations of En-Cs.

| Source | French troops have left their area of responsibility in Afghanistan ( Kapisa and Surobi ) . |
|---|---|
| Reference | Francouzské jednotky opustily svou oblast odpovědnosti v Afghánistánu ( Kapisa a Surobi ) . |
| DCNMT | Francouzské jednotky opustily svou oblast odpovědnosti v Afghánistánu ( Kapisa a Surois ) . |
| Source | " All the guests were made to feel important and loved " recalls the top model , who started working with him during Haute Couture Week Paris , in 1995 . |
| Reference | Všichni pozvaní se díky němu mohli cítit důležití a milovaní , " vzpomíná top modelka , která s ním začala pracovat v průběhu Pařížského týdne vrcholné módy v roce 1995 . |
| DCNMT | " Všichni hosté byli provedeni , aby se cítili důležití a milovaní " připomíná nejvyšší model , který s ním začal pracovat v průběhu týdeníku Haute Coutupe v Paříži v roce 1995 . |
| Source | " There are so many private weapons factories now , which do not endure competition on the international market and throw weapons from under the counter to the black market , including in Moscow , " says the expert . |
| Reference | " V současnosti vznikají soukromé zbrojařské podniky , které nejsou konkurenceschopné na mezinárodním trhu , a vyřazují zbraně , které dodávají na černý trh včetně Moskvy , " říká tento odborník . |
| DCNMT | " V současnosti existuje tolik soukromých zbraní , které nevydrží hospodářskou soutěž na mezinárodním trhu a hodí zbraně pod pultem k černému trhu , včetně Moskvy , " říká odborník . |

Table 5: Sample translations of Cs-En.

| Source | Prezident Karzáí nechce zahraniční kontroly , zejména ne při příležitosti voleb plánovaných na duben 2014 . |
|---|---|
| Reference | President Karzai does not want any foreign controls , particularly on the occasion of the elections in April 2014 . |
| DCNMT | President Karzai does not want foreign controls , particularly in the opportunity of elections planned on April 2014 . |
| Source | Manželský pár měl dvě děti , Prestona a Heidi , a dlouhou dobu žili v kalifornském městě Malibu , kde pobývá mnoho celebrit . |
| Reference | The couple had two sons , Preston and Heidi , and lived for a long time in the Californian city Malibu , home to many celebrities . |
| DCNMT | The married couple had two children , Preston and Heidi , and long lived in the California city of Malibu , where many celebrities resided . |
| Source | Trestný čin rouhání je zachován a urážka je nadále zakázána , což by mohlo mít vážné důsledky pro svobodu vyjadřování , zejména pak pro tisk . |
| Reference | The offence of blasphemy is maintained and insults are now prohibited , which could have serious consequences on freedom of expression , particularly for the press . |
| DCNMT | The criminal action of blasphemy is maintained and insult is still prohibited , which could have serious consequences for freedom of expression , especially for the press . |

