# Peer review of "Deep Character-Level Neural Machine Translation By Learning Morphology"

_ICLR 2017 — rejected_

[Official Review · AnonReviewer3 · rating 5 · confidence 5 · 16 Dec 2016 (modified: 22 Dec 2016)]
**Well-executed paper with good analysis but little novelty**

Update after reading the authors' responses & the paper revision dated Dec 21:
I have removed the comment "insufficient comparison to past work" in the title & update the score from 3 -> 5.
The main reason for the score is on novelty. The proposal of HGRU & the use of the R matrix are basically just to achieve the effect of "whether to continue from character-level states or using word-level states". It seems that these solutions are specific to symbolic frameworks like Theano (which the authors used) and TensorFlow. This, however, is not a problem for languages like Matlab (which Luong & Manning used) or Torch.

-----

This is a well-written paper with good analysis in which I especially like Figure 5. However I think there is little novelty in this work. The title is about learning morphology but there is nothing specifically enforced in the model to learn morphemes or subword units. For example, maybe some constraints can be put on the weights in w_i in Figure 1 to detect morpheme boundaries or some additional objective like MDL can be used (though it's not clear how these constraints can be incorporated cleanly). 

Moreover, I'm very surprised that litte comparison (only a brief mention) was given to the work of (Luong & Manning, 2016) [1], which trains deep 8-layer word-character models and achieves much better results on English-Czech, e.g., 19.6 BLEU compared to 17.0 BLEU achieved in the paper. I think the HGRU thing is over-complicated in terms of presentation. If I read correctly, what HGRU does is basically either continue the character decoder or reset using word-level states at boundaries, which is what was done in [1]. Luong & Manning (2016) even make it more efficient by not having to decode all target words at the morpheme level & it would be good to know the speed of the model proposed in this ICLR submission. What end up new in this paper are perhaps different analyses on what a character-based model learns & adding an additional RNN layer in the encoder.

One minor comment: annotate h_t in Figure 1.

[1] Minh-Thang Luong and Christopher D. Manning. 2016. Achieving Open Vocabulary Neural Machine Translation
with Hybrid Word-Character Models. ACL.

[Official Review · AnonReviewer2 · rating 7 · confidence 4 · 16 Dec 2016]
**Good paper, accept**

The paper presents one of the first neural translation systems that operates purely at the character-level, another one being

[Author Response · Shenjian Zhao · 17 Dec 2016]
**Training time**

Dear reviewers,

We uploaded a slightly modified version. We have added the training time for each model to Table 1 and clarified some notations.

Thanks!

[Author Response · Shenjian Zhao · 18 Dec 2016]
**Adding several comparisons**

Dear reviewers,

We uploaded a slightly modified version. 
We have further explained the novelty of the hierarchical decoder in Section 3.2.
We have added the comparison with Luong & Manning, 2016 [1] and the trivial baseline (CNMT) to Table 1.
The trivial baseline is the old version of this submission which takes the last hidden state of RNN as the representation of the source word. You could find it on arxiv (

[Official Review · AnonReviewer1 · rating 6 · confidence 4 · 20 Dec 2016]
**A well written paper**

* Summary: This paper proposes a neural machine translation model that translates the source and the target texts in an end to end manner from characters to characters. The model can learn morphology in the encoder and in the decoder the authors use a hierarchical decoder. Authors provide very compelling results on various bilingual corpora for different language pairs. The paper is well-written, the results are competitive compared to other baselines in the literature.


* Review:
     - I think the paper is very well written, I like the analysis presented in this paper. It is clean and precise. 
     - The idea of using hierarchical decoders have been explored before, e.g. [1]. Can you cite those papers?
     - This paper is mainly an application paper and it is mainly the application of several existing components on the character-level NMT tasks. In this sense, it is good that authors made their codes available online. However, the contributions from the general ML point of view is still limited.
                   
* Some Requests:
 -Can you add the size of the models to the Table 1? 
- Can you add some of the failure cases of your model, where the model failed to translate correctly?

* An Overview of the Review:

Pros:
    - The paper is well written
    - Extensive analysis of the model on various language pairs
    - Convincing experimental results.    
    
Cons:
    - The model is complicated.
    - Mainly an architecture engineering/application paper(bringing together various well-known techniques), not much novelty.
    - The proposed model is potentially slower than the regular models since it needs to operate over the characters instead of the words and uses several RNNs.

[1] Serban IV, Sordoni A, Bengio Y, Courville A, Pineau J. Hierarchical neural network generative models for movie dialogues. arXiv preprint arXiv:1507.04808. 2015 Jul 17.

[Author Response · Shenjian Zhao · 22 Dec 2016]
**Adding an appendix and model size comparison**

Dear reviewers,

We uploaded an updated version with an appendix.  In the appendix, we describe our model in detail and add more translation samples. 
We have added the size of models in Table 1. Table 1 becomes more comprehensive, thanks for your insightful suggestions.

Thanks.

[Final Decision · Program Chairs · 06 Feb 2017]
**ICLR committee final decision**

This paper is concurrently one of the first successful attempts to do machine translation using a character-character MT modeling, and generally the authors liked the approach. However, the reviewers raised several issues with the novelty and experimental setup of the work. 
 
 Pros:
 - The analysis of the work was strong. This illustrated the underlying property, and all reviewers praised these figures.
 
 Mixed: 
 - Some found the paper clear, praising it as a "well-written paper", however other found that important details were lacking and the notation was improperly overloaded. As the reviewers were generally experts in the area, this should be improved
 - Reviewers were also split on results. Some found the results quite "compelling" and comprehensive, but others thought there should be more comparison to BPE and other morphogically based work
 - Modeling novelty was also questionable. Reviewers like the partial novelty of the character based approach, but felt like the ML contributions were too shallow for ICLR
 
 Cons:
 - Reviewers generally found the model itself to be overly complicated and the paper to focus too much on engineering. 
 - There were questions about experimental setup. In particular, a call for more speed numbers and broader comparison.